

# Weak vs. strong breaking of integrability in interacting scalar quantum field theories

Bence Fitos[1,2] and Gábor Takács[1,2,3]

**1** Department of Theoretical Physics, Institute of Physics, Budapest University of Technology and Economics, Műegyetem rkp. 3., H-1111 Budapest, Hungary
**2** BME-MTA Statistical Field Theory 'Lendület' Research Group, Budapest University of Technology and Economics, Műegyetem rkp. 3., H-1111 Budapest, Hungary
**3** MTA-BME Quantum Dynamics and Correlations Research Group, Budapest University of Technology and Economics, Műegyetem rkp. 3., H-1111 Budapest, Hungary

## Abstract

The recently proposed classification of integrability-breaking perturbations according to their strength is studied in the context of quantum field theories. Using random matrix methods to diagnose the resulting quantum chaotic behaviour, we investigate the $\phi^4$ and $\phi^6$ interactions of a massive scalar, by considering the crossover between Poissonian and Wigner-Dyson distributions in systems truncated to a finite-dimensional Hilbert space. We find that a naive extension of the scaling of crossover coupling with the volume observed in spin chains does not give satisfactory results for quantum field theory. Instead, we demonstrate that considering the scaling of the crossover coupling with the number of particles yields robust signatures, and is able to distinguish between the strengths of integrability breaking in the $\phi^4$ and $\phi^6$ quantum field theories.

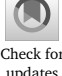

# 1 Introduction

Integrability is fundamental to our understanding of the dynamics of quantum many-body systems, with its breaking related to quantum chaos, ergodicity, and thermalisation [1–3]. In this work we consider integrability breaking in relativistic quantum field theories which play a fundamental role in describing universality classes of quantum systems near critical points.

Integrability is related to the presence of higher conserved charges, which generally make the number of (quasi)particle excitations conserved as they restrict the set of incoming and outgoing momenta to be the same in any multi-particle scattering process and also restrict all amplitudes to the product of independent two-particle amplitudes [4,5]. Integrability breaking perturbations lead to violations of the higher conservation laws, with weak integrability breaking being characterised by breaking them at higher order in the coupling parameter [6–11].

Recent developments show that integrability breaking can be classified by its strength characterised by the order in which the higher conservation laws are broken. One approach identified weak integrability breaking by studying thermalisation in interacting scalar field theory using Boltzmann kinetic equation [12]. On the other hand, studying integrability preserving deformations of integrable spin chains [6,7] revealed that the strength of integrability breaking is also manifested in the crossover of level spacing statistics [9,13]. The latter approach eventually leads to a hierarchy of deformations integrability characterised by the order of the perturbation at which integrability breaking happens.

Integrability breaking can be studied using tools from random matrix theory which provides a paradigmatic approach to quantum chaos [14,15] and recently there has been a renewed interest in applying it to quantum many-body models [16,17], and to quantum field theories [18–20]. For spin chains of finite length $L$, integrability breaking results in a crossover between Poisson and Wigner-Dyson forms of level spacing statistics [21]. Previous works found that for interacting systems the crossover coupling generally scales with $L$ as $1/L^3$ [22,23], which was confirmed in [9] for the gapless regime of the XXZ spin chain. For weak integrability breaking the crossover was found to be significantly slower [9,13]: in the gapless regime of the XXZ spin chain, the behaviour was observed to be $1/L^2$ [9], while Ref. [13] found $1/L$ for a weakly chaotic perturbation of the XXX spin chain. In the gapped regime of the XXZ chain, the crossover coupling was found to decay exponentially with the volume, again with a significantly slower decay for weak breaking of integrability compared to the strong case [9]. Therefore the scaling of the crossover coupling with system size appears to be a good indicator of the strong/weak nature of integrability breaking.

In $1+1$-dimensonal quantum field theory, two-particle $\rightarrow$ two-particle scattering processes are kinematically constrained to have the same set of incoming and outgoing momenta. As a result, it can be argued that $\phi^4$ theory only violates integrability at the second order in the coupling, while $\phi^6$ theory is expected to violate it at the first order, which is supported by a Boltzmann equation approach to their non-equilibrium dynamics [12]. The full quantum non-equilibrium dynamics of the $\phi^4$ model was recently studied in [24], where only a very slow relaxation was found with no equilibration on the available time scales, consistent with the suggestion that $\phi^4$ leads to a weak breaking of integrability.

It is then a natural problem to ask whether, in analogy with the spin chains, the different strengths of integrability breaking are manifested in the scaling of the crossover in the level spacing statistics with the system size. In this work, we investigate this issue in interacting scalar field theory, where the integrability of a massive free field theory is broken by a $\phi^n$ interaction term. We adopt the truncated Hamiltonian approach (THA) which has already been successfully applied to determine level-spacing statistics in perturbations of the tricritical Ising model [25] and also in other field theories including the $\phi^4$ model [12]; in our study, we go substantially further by studying the scaling of the crossover with the system size.

The outline of the paper is as follows. In Section 2 we outline the renormalisation group improved version of the THA and specify the relevant quantities extracted from the level spacing statistics that can be used to quantify the crossover between integrable and non-integrable behaviour. In Section 3 we study the behaviour of the crossover as a function of the volume, and in Section 4 we analyse it in terms of the number of particle excitations, presenting our conclusions in Section 5. Details concerning renormalisation group improvements in THA are relegated to Appendix A.

## 2 THA and level spacing statistics

### 2.1 The RG improved truncated Hamiltonian approach

We consider massive scalar field theories in $1+1$ dimensions. The free theory is described by the Klein–Gordon Hamiltonian

$$H_{\text{KG}} = \frac{1}{2} \int dx : \left((\partial_t \phi)^2 + (\partial_x \phi)^2 + m^2 \phi^2\right) :, \tag{1}$$

where $: \dots :$ denotes normal ordering concerning the free modes. The interaction term is defined as

$$V_n = \int dx : \phi^n(x) : . \tag{2}$$

We consider the cases $n = 4$ and $6$, corresponding to the so-called $\phi^4$ and $\phi^6$ theories:

$$H_{\phi^4} = H_{\text{KG}} + g_4 V_4 , \tag{3}$$

$$H_{\phi^6} = H_{\text{KG}} + g_6 V_6 , \tag{4}$$

in a finite volume $0 \le x \le L$ with periodic boundary conditions $\phi(L) = \phi(0)$. The models can be specified by the dimensionless volume $mL$ and the $\tilde{g}_n = g_n/m^2$ dimensionless coupling parameters.

To evaluate the spectrum of the theory we use the truncated Hamiltonian approach, pioneered by Yurov and Zamolodchikov [26, 27] and later extended to the $\phi^4$ model [28–32]. The method consists of constructing the Hamiltonian as a matrix in the computational basis formed by the eigenstates of the free massive theory (1), which is discrete in finite volume, and introducing an upper cutoff $\Lambda$ on the energy of the states retained, i.e. restricting the Hilbert space to states $|k\rangle$ satisfying $H_{\text{KG}} |k\rangle = E_k |k\rangle$ with $E_k \le \Lambda$.

The truncation procedure results in an approximation to the exact spectrum, with truncation errors dependent on the cutoff $\Lambda$. In any given energy window, the truncated spectrum is expected to converge to the exact one when the cutoff is increased, due to the $V_n$'s being (strongly) relevant operators in the renormalisation group (RG) sense. Eventually, the leading order cutoff dependence can be eliminated by RG methods [33–35]. Here we follow the approach introduced in [30], which for a Hamiltonian

$$H = H_{\text{KG}} + \sum_n g_n V_n , \tag{5}$$

leads to the following renormalisation group improvement at leading order in the cutoff:

$$\Delta H = \sum_n \kappa_n V_n , \quad \text{where} \quad \kappa_k = -\sum_{n \ge m} g_n g_m \int_\Lambda^\infty dE \frac{\mu_{nmk}(E)}{E - \mathcal{E}} . \tag{6}$$

The $\mu_{nmk}(E)$ functions describe the running of the couplings with the cutoff $\Lambda$, and are summarized in Eq. (A.20). The reference energy $\mathcal{E}$ must be set in the range of the energy levels that the method is optimised for. For more details on the RG improvement the reader is referred to Appendix A. The counter terms for the $\phi^4$ model were obtained in [30], while the terms involving the interaction $\phi^6$ are a new result of the present work.

## 2.2 Level spacing statistics from THA

Integrability breaking can be investigated using the level spacing statistics constructed from the normalized level spacing $s_n = (E_{n+1} - E_n)\omega(E_n)$, where $E_n$ is the $n$-th energy level, and $\omega(E_n)$ is the smoothed level density around $E_n$. The Hamiltonians considered here are real and symmetric, therefore the level spacing statistics is expected to follow the Wigner-Dyson distribution for the Gaussian orthogonal ensemble

$$P_{\text{GOE}}(s) = \frac{\pi}{2} s \exp\left(-\frac{\pi}{4} s^2\right), \tag{7}$$

when integrability is broken, while for the integrable limit, it is expected to be Poissonian

$$P_{\text{P}}(s) = \exp(-s). \tag{8}$$

For a system truncated to a finite-dimensional Hilbert space, the level spacing distribution is a continuous function of the coupling, with the transition becoming sharper as the dimension of the Hilbert space is increased [21, 25]. In addition, due to the locality of the Hamiltonian the level spacing from the full spectrum is found to deviate from the random matrix prediction because of the structure of low-lying levels dictated by quasi-particle excitations. This problem can be solved by constructing the level spacing distribution from the middle part of the spectrum staying sufficiently away from the low-energy states and also from the truncation scale $\Lambda$. Therefore, we aim to determine the energy spectrum in some energy window $[E_1, E_2]$ which is ideally chosen so that $m \ll E_1 < E_2 \ll \Lambda$. To optimise the precision of the THA spectrum we set the reference energy as $\mathcal{E} = (E_1 + E_2)/2$, and push the cutoff energy $\Lambda$ as high as possible while still keeping the computing time within reasonable bounds. Therefore, the computed energy spectrum and therefore the level spacing distribution depends on the dimensionless parameters $\{mL, \tilde{g}_n = g_n/m^2, \Lambda/m, E_1/m, E_2/m\}$.

Furthermore, it is also necessary to avoid trivial degeneracies due to global symmetries such as

- translational invariance: $x \to x + a$,

- parity: $x \to -x$, and

- $\mathbb{Z}_2$ symmetry: $\phi \to -\phi$.

This is achieved by restricting to states of zero total momentum, which are also even under both parity and $\mathbb{Z}_2$ symmetry, i.e., to the sector of the Hilbert space that contains the vacuum.

## 2.3 Quantifying integrability breaking

We choose the following measures to follow the crossover of the level spacing distribution from Poisson to GOE statistics quantitatively:

(1) *Consecutive level spacing ratios* – The consecutive level ratios are defined as [18, 36]

$$r_n = \frac{s_n}{s_{n-1}}, \qquad \tilde{r}_n = \min(r_n, 1/r_n). \tag{9}$$

The average value of this variable is $\langle \tilde{r}_{\mathrm{Poi}} \rangle = 2\log 2 - 1 \approx 0.386$ for Poissonian, and $\langle \tilde{r}_{\mathrm{GOE}} \rangle = 4 - 2\sqrt{3} \approx 0.534$ for GOE statistics, respectively. We consider an affine map from the interval $[\langle \tilde{r}_{\mathrm{Poi}} \rangle, \langle \tilde{r}_{\mathrm{GOE}} \rangle]$ to $[0, 1]$:

$$\langle \tilde{r}' \rangle = \frac{\tilde{r} - \langle \tilde{r}_{\mathrm{Poi}} \rangle}{\langle \tilde{r}_{\mathrm{GOE}} \rangle - \langle \tilde{r}_{\mathrm{Poi}} \rangle}, \tag{10}$$

which takes value 0 for Poissonian and 1 for GOE statistics. Since this is a statistical measure, its uncertainty can be estimated using the central limit theorem and the known variances $\delta(\tilde{r}_{\mathrm{Poi}}) \approx 0.280$ and $\delta(\tilde{r}_{\mathrm{GOE}})) \approx 0.254$ in the two ensembles, leading to an upper estimate $\delta \langle \tilde{r}' \rangle \lesssim 0.280/\sqrt{N}$ where $N$ is the number of levels included in the statistics.

(2) *Brody distribution* – The Brody distribution [22]

$$P_\beta(s) = (\beta + 1) b s^\beta \exp\left(-b s^{\beta+1}\right), \tag{11}$$

with

$$b = \Gamma\left(\frac{\beta + 2}{\beta + 1}\right)^{\beta+1}, \tag{12}$$

can be used to interpolate between the Poissonian and GOE cases, with $\beta = 0$ corresponding to Poisson, while $\beta = 1$ corresponds to GOE statistics. Given a set of energy levels spectrum, it can be evaluated by making a histogram from the $s_n$ spacings with some appropriate resolution and then fitting $P_\beta(s)$ to extract $\beta$, as illustrated in Figure 1. Statistical fluctuations can be estimated by assigning to every histogram value its square root as an estimator for its variance, and then computing the variance $\delta\beta$ of the fitted value of $\beta$ using the standard least squares method.

## 2.4 Crossover coupling

Consider now the Hamiltonians of the $\phi^n$ theories for $n = 4$ and 6, and follow the behaviour of the level spacing statistics for weak coupling $g_n$. When $g_n = 0$, the theory is integrable and therefore one expects Poissonian statistics, however, when turning on a non-zero coupling the dynamics is expected to become chaotic implying GOE statistics, with a continuous crossover between the two behaviours due to the finiteness of the energy window and the volume.

Figure 2 shows the variation of the measures $\langle \tilde{r}' \rangle$ and $\beta$ in the $\phi^6$ theory with increasing coupling $g_6$ while keeping all other parameters such as the volume, the cutoff, and the energy window fixed. Note that both parameters show a continuous crossover between their Poissonian (0) and GOE (1) values. Motivated by previous studies performed with spin chains [23] we fit the data with an exponential relaxation

$$f_{\tilde{g}^*}(\tilde{g}) = 1 - \exp\left(-\tilde{g}/\tilde{g}^*\right), \tag{13}$$

to determine the *crossover coupling* $\tilde{g}^*$. Note that the eventual value of $\tilde{g}^*$ depends on the measure used to define it; however, this is not surprising given the continuous nature of the crossover.

We remark that the spectrum statistics in the free theory (i.e., $g_n = 0$) is not eventually Poissonian due to a higher number of degeneracies in free field theory than the one expected from integrability alone. However, turning on the interaction, these degeneracies are instantly resolved and the spectrum quickly becomes Poissonian, much more rapidly than the eventual crossover to GOE,[1] therefore Poissonian statistics is observed for very small but non-zero couplings (i.e., $0 \neq g_n \ll g_n^*$, c.f. Fig. 2).

---

[1] Except for very small volumes $mL \lesssim 2$.

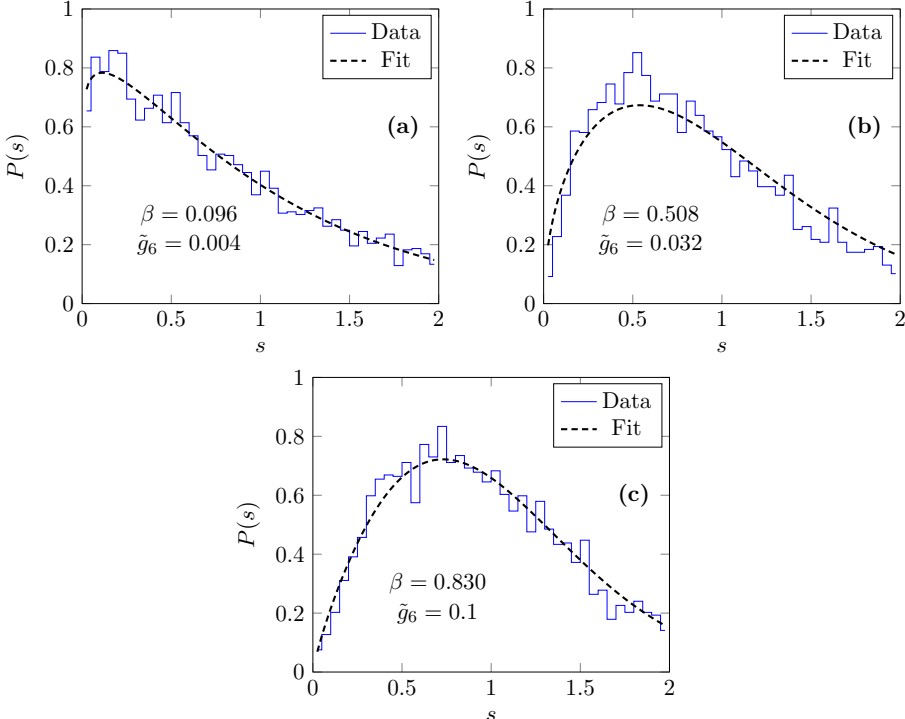

Figure 1: Level spacing histograms (*Data*) and the corresponding Brody distributions (*Fit*) in $\phi^6$ theory at different values of the coupling, with the extracted value of $\beta$ shown as part of the legend. The parameters of the spectral window are $mL = 12, [E_1/m, E_2/m] = [12, 16], \Lambda/m = 20$.

## 3 Finite size scaling

As discussed above, the crossover coupling is expected to go to zero with increasing system size. Previous studies in spin chains suggest considering the behaviour of the crossover coupling $\tilde{g}^*$ as a function of the volume $L$. Based on the exponential behaviour found for the massive regime of the XXZ spin chain [9], we fit the volume dependence $\tilde{g}^*(mL)$ as

$$\log(\tilde{g}^*) = a + b(mL). \tag{14}$$

Note that redefining the normalisation of the coupling constant affects only $a$, therefore the eventual quantity of interest is the coefficient $b$, which can be taken to characterize the strength of integrability breaking of the given interaction ($V_n$) [9].

Figure 3 illustrates the $\log(\tilde{g}^*)$ vs. $mL$ data for the $\phi^4$ field theory using consecutive level ratios and Brody distribution. Unfortunately, the volume range is quite restricted: on the one hand, the number of energy levels increases roughly exponentially as a function of both the volume and the cutoff energy, while on the other hand, the energy window needs to be wide enough to contain enough levels for statistical analysis. Different ranges of volume can be accessed by varying the energy window setup. The results obtained for $\phi^4$ and $\phi^6$ theories are summarized in Table 1.

Note that the $b$ values corresponding to different measures agree quite well (within the estimated statistical errors), despite the visible difference between the individual values $\tilde{g}^{\beta*}$ and $\tilde{g}^{\tilde{r}*}$ of the crossover couplings (c.f. Fig. 3). This is encouraging as it indicates that the scaling of the crossover coupling with system size is universal and thus physically meaningful. However, the $b$ values depend substantially on the choice of the energy window and the cutoff. In addition, the crossover couplings vary only by a small amount, in stark contrast to

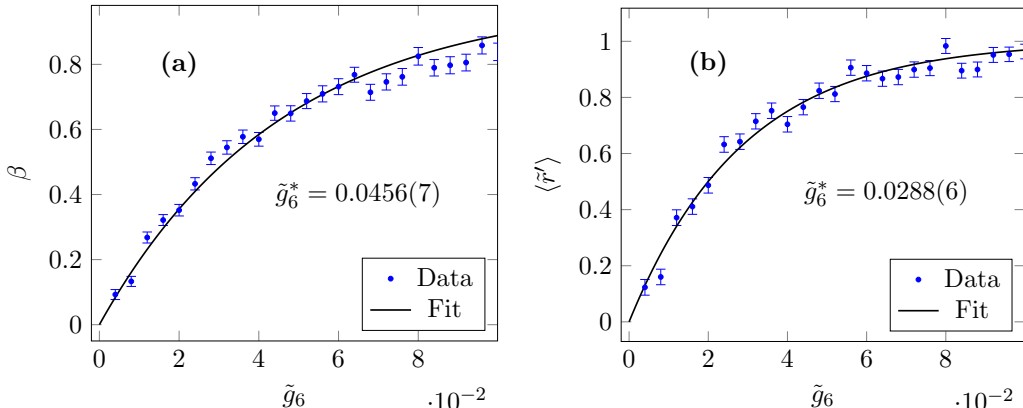

Figure 2: The $\beta$ and $\langle\tilde{r}'\rangle$ measures as a function of the $g_6$ coupling in the $\phi^6$ theory. We also present the fitted curve defined in Eq. (13) determining the coupling constant. The error bars ($1\sigma$) and the error of the crossover coupling are calculated from the statistical uncertainty. We note that $\langle\tilde{r}'\rangle(g_6 = 0) = -1.35$ and $\beta(g_6 = 0) = -2.00$ due to additional symmetries in the free theory. Parameters: $mL = 12, [E_1/m, E_2/m] = [12, 16], \Lambda/m = 20$.

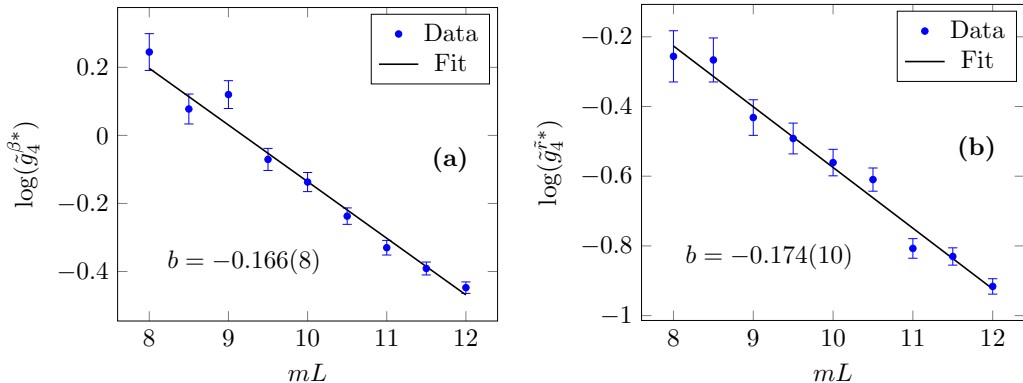

Figure 3: The $\log(\tilde{g}_4^*)$ vs. $mL$ graphs in the $\phi^4$ thoery using the (a) $\beta$ and the (b) $\langle\tilde{r}'\rangle$ measures. We also depicted the fitted line according to Eq. (14). The errors are calculated from the statistical uncertainty. Parameters: $[E_1/m, E_2/m] = [12, 16], \Lambda/m = 20$.

the results obtained for spin chains [9], which calls into question the physical significance of the results. As a result, the volume dependence of the crossover coupling allows no reliable conclusions concerning the difference in the strengths of integrability breaking of the $\phi^4$ and $\phi^6$ perturbations.

## 4 Scaling in particle number

An alternative notion of system size is to consider energy levels with a given number of particles present. For spin chains, their length $L$ is in fact the same parameter since the Hilbert space is dominated by states where the number of quasi-particle excitations of order $L$. However, particle number is only a good quantum number for integrable systems with higher conserved charges and consequently factorised scattering, so it is not immediately apparent that this approach should work.

Table 1: Finite size scaling of the crossover coupling in the $\phi^4$ and $\phi^6$ theories using $\beta$ and $\langle \tilde{r}' \rangle$ measures at different energy windows and cutoff energies. The error in the brackets is calculated from the statistical uncertainty. The examined volume regimes at the corresponding energy windows: $[12.0, 16.0]: mL = 8, 8.5, ..., 12$; $[8.0, 10.7]: mL = 16, 17, ..., 23$; $[6.5, 9.1]: mL = 24, 24.5, ..., 30$.

| $[E_1/m, E_2/m]$ | $\Lambda/m$ | $b_{\phi^4}^{\beta}$ | $b_{\phi^4}^{\tilde{r}}$ | $b_{\phi^6}^{\beta}$ | $b_{\phi^6}^{\tilde{r}}$ |
|---|---|---|---|---|---|
| $[12.0, 16.0]$ | 20.0 | -0.166(8) | -0.174(10) | -0.215(6) | -0.225(8) |
| $[8.0, 10.7]$ | 14.0 | -0.058(4) | -0.059(6) | -0.076(3) | -0.080(5) |
| $[8.0, 10.7]$ | 13.0 | -0.071(4) | -0.069(6) | -0.077(3) | -0.079(5) |
| $[6.5, 9.1]$ | 11.5 | -0.053(4) | -0.048(5) | -0.046(3) | -0.053(4) |
| $[6.5, 9.1]$ | 10.5 | -0.053(4) | -0.070(5) | -0.049(3) | -0.051(4) |

For the theories considered here, it is natural to classify the states of the computational basis by the eigenvalue of the free boson particle number operator $\hat{N}$. Noting that the crossover coupling is expected to go to zero with increasing system size, we consider integrability-breaking processes in the weak coupling limit. As we already discussed, the simplest processes correspond to $2 \rightarrow 2$ scattering, which always has the same set of incoming and outgoing momenta.

Integrability-breaking multi-particle processes can be classified by their property concerning particle number $N$. To change the level spacing statistics, processes that preserve particle numbers must still change the set of particle momenta and lift level crossings between states with the same value of $\hat{N}$. In the $\phi^4$ and $\phi^6$ models considered here, the lowest order such processes correspond to $3 \rightarrow 3$ scattering. Particle number changing processes such as $2 \rightarrow 4$ lift level crossings between levels in subspaces with different values of $\hat{N}$. However, such processes are kinematically suppressed by the requirement of threshold energy. The opposite process $4 \rightarrow 2$ has no threshold but is suppressed compared to $3 \rightarrow 3$ due to the requirement of four particles meeting simultaneously (i.e., within the finite range of interaction imposed by the mass gap). As a result, one expects that the level spacing statistics restricted to subspaces with fixed $N$ show a much faster crossover than the statistics for the overall spectrum. While these considerations are quite heuristic, the results obtained below are consistent with them.

Therefore we consider the interacting Hamiltonians (3) and (4) within individual subspaces of a given number $N$ of particles, i.e., we neglect the interaction between the different $N$-sectors and consider them individually. To get an idea of whether the number of states with a given number of particles is sufficient for statistical analysis, one can determine the cardinality of the computational basis for given values of the volume and the cutoff as a function of $N$, which is illustrated in Figure 4. We then perform the aforementioned statistical analysis for the spectra corresponding to an individual $N$-sector as we switch on the interaction, from which we determine the $g^*(N)$ crossover coupling.

Figure 5 shows the crossover couplings corresponding to some $N$-sectors in the $\phi^4$ and $\phi^6$ theory at $mL = 6$ volume. We observe that the crossover couplings cover a remarkably wider range on the log scale than in the volume scaling case illustrated in Fig. 3. In addition, the crossover couplings become especially small with increasing $N$, which shows that our assumptions are self-consistent. For comparison, we also determined the crossover coupling using the complete spectrum at $mL = 6$ as in Section 3, including particle-changing interaction.[2] The crossover couplings corresponding to the full spectra (at the same volume, energy window,

---

[2]Note that it does not make sense to compute statistics for the full spectrum with neglected particle-changing interaction as the enforced independence of the different $N$-sectors always results in almost Poissonian statistics.

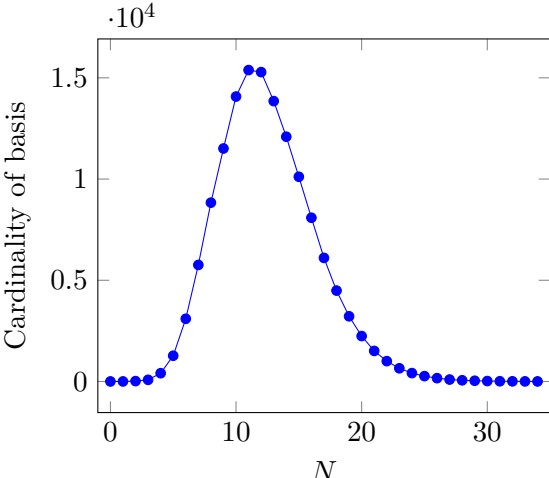

Figure 4: Cardinality of the THA basis as a function of the particle number. Parameters: $mL = 6, \Lambda/m = 35$.

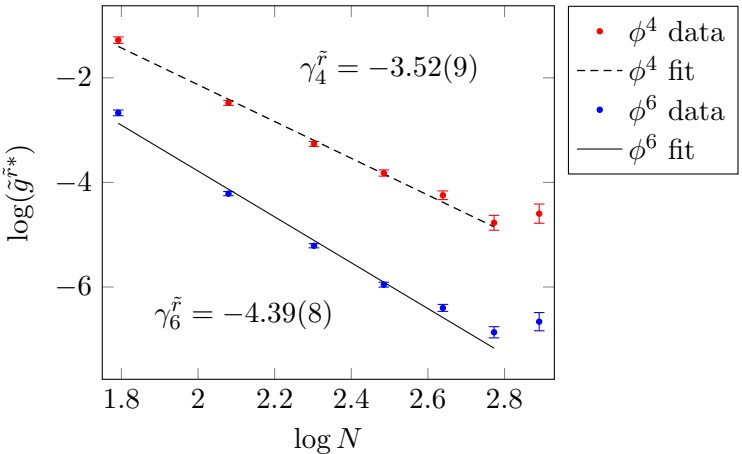

Figure 5: Crossover coupling vs. particle number ($N = 6, 8, ..., 18$) in the $\phi^4$ and $\phi^6$ theories, together with linear fits according to Eq. (16) (the $N = 18$ points were ignored). The errors are calculated from statistical uncertainty. Parameters: $mL = 6$, $[E_1/m, E_2/m] = [20, 25]$, and $\Lambda/m = 35$.

and cutoff as in Fig. 5) are found to be

$$\log \tilde{g}_4^* = -0.72(3), \quad \text{and} \quad \log \tilde{g}_6^* = -2.86(2). \tag{15}$$

Comparing to Fig. 5 we find that the typical value of $\tilde{g}^*(N)$ is significantly smaller than the crossover coupling corresponding to the full spectrum, supporting the argument that the level crossings within the $N$-sectors are lifted much faster than those occurring between states corresponding to different $N$.

Although we have no particular argument for a power scaling of $\tilde{g}^*(N)$ vs. $N$, Fig. 5 strongly suggests applying a linear fit to the $\log g^*$ vs. $\log N$ data. We define the exponent $\gamma$ by assuming the dependence

$$\log g^* = \gamma \log N + \alpha, \tag{16}$$

i.e. $g^* \sim N^\gamma$. Similarly to the finite volume scaling parameter $b$ defined in (14), the exponent $\gamma$ is independent of normalisation of the coupling. The fitted lines are illustrated in Figure 5;

Table 2: Scaling parameter characterizing the crossover coupling vs. particle number dependence in the $\phi^4$ and $\phi^6$ theories using $\beta$ and $\langle \tilde{r}' \rangle$ measures at different volumes, energy windows, and cutoff energies. The uncertainty is calculated from the statistical fluctuations.

| $mL$ | $[E_1/m, E_2/m]$ | $\Lambda/m$ | $\gamma_4^\beta$ | $\gamma_4^{\tilde{r}}$ | $\gamma_6^\beta$ | $\gamma_6^{\tilde{r}}$ |
|---|---|---|---|---|---|---|
| 4 | [30, 40] | 50 | -3.15(3) | -3.00(4) | -4.50(3) | -4.21(4) |
| 5 | [29, 35] | 45 | -3.14(3) | -3.19(4) | -4.44(3) | -4.45(3) |
| 6 | [20, 25] | 35 | -3.35(6) | -3.52(9) | -4.34(6) | -4.39(8) |
| 7 | [20, 25] | 35 | -3.45(4) | -3.57(6) | -4.76(4) | -4.84(5) |
| 8 | [17, 22] | 28 | -3.62(5) | -3.70(8) | -4.80(4) | -4.76(6) |

note that the $N = 18$ points were ignored as the relevant energy levels are too close to the edge of the energy window.

We repeated the computations of the exponent $\gamma$ at different volumes, with the results summarized in Table 2. Now the choice of the parameters is less constrained than for the analysis of the full spectrum, as the number of states in an $N$-sector only increases polynomially with the cutoff. Nevertheless, it is still necessary to change the energy window and the cutoff as the volume is increased.

The results demonstrate some key features:

1. Similarly to the $b$ values obtained from finite volume scaling, the $\gamma$ values coming from the Brody distribution and the consecutive level ratios match pretty well, indicating that the exponent does capture universal features that only depend on the interaction.

2. $\gamma$ values only change by about 10-20% as the volume is doubled, which makes their value much more consistent than the $b$ values for finite volume scaling (c.f. Table 1).

3. Comparing the $\gamma$ values for $\phi^4$ and $\phi^6$ perturbations, the $\gamma_4$ are significantly smaller than the $\gamma_6$, which can be interpreted as a clear signal that the $\phi^6$ interaction violates integrability stronger than the $\phi^4$.

This leads us to the conclusion that the relevant system size parameter for the crossover from integrable to non-integrable level spacing statistics is the particle number rather than the volume. As noted before, this eventually does not contradict previous results obtained in spin chains, since in those systems the two parameters are closely related when considering the middle of the spectrum used in the analysis of level spacing statistics.

## 5 Conclusion

In this work, we investigated integrability breaking in the $\phi^4$ and $\phi^6$ theory by analyzing the statistical properties of the energy spectrum. We evaluated level spacing statistics using the truncated Hamiltonian approach and used it to analyse the crossover from Poisson distribution characteristic of integrability to the Wigner-Dyson distribution resulting from the Gaussian orthogonal ensemble of random matrices, which corresponds to quantum chaotic systems with a real symmetric Hamiltonian. Previous studies showed that the dependence of the crossover coupling on the system size can be used to characterise the strength of integrability breaking.

The first main conclusion of our study is that the relevant parameter of system size in which to consider the scaling of the crossover coupling is the number of particle excitations rather than the volume. The volume dependence of the crossover coupling turned out to be

particularly weak. On the contrary, we found a much stronger scaling behavior by analyzing its dependence on the particle number. We found that the scaling exponent showed good agreement between two different determinations of the crossover coupling, which is an important consistency check for the approach.

The rapid scaling of the crossover coupling in $N$ indicates that the relevant parameter (in terms of integrability breaking) is the particle number, not the volume. We also found that the crossover couplings corresponding to higher $N$'s are typically well below the one corresponding to the total spectrum, which means that the dominant integrability-breaking processes are those preserving the number of particles.

This observation can be intuitively related to (classical) Hamiltonian dynamics, where the KAM theorem [37–39] asserts that for weak perturbations of an integrable with a finite number of degrees of freedom, the dynamics does not immediately become fully chaotic: a portion of the phase space retains the structure of the tori characteristic of integrability, albeit their shape is deformed. The threshold for the breakup of such tori, however, approaches zero in the limit when the number of degrees of freedom goes to infinity [40, 41]. This corresponds to a smooth crossover that becomes progressively sharper when the number of degrees of freedom is increased, eventually transitioning to a non-integrable behaviour for any small value of the integrability breaking coupling in the thermodynamic limit. We note that it is the number of degrees of freedom, encoded in our case in the number of particles, which is the natural control parameter for the transition. We also reiterate that this observation does not in any way contradict the scaling of the crossover with the length of the system observed in spin chains, since for the relevant states (those in the middle of the spectrum) the length of the chain and the number of quasi-particle excitations are essentially the same parameters.

Our second main conclusion is that the scaling of the crossover coupling, considered as a function of particle number supports the distinction proposed in [12] on the basis of non-equilibrium evolution modeled using a Boltzmann kinetic equation approach, according to which the $\phi^4$ induces weak integrability breaking as opposed to the strong one induced by $\phi^6$. An interesting open issue is to compare the full quantum evolution in the $\phi^4$ and $\phi^6$ theories e.g. following [24]; we performed some preliminary studies which indicated that the THA must be improved further to arrive at reliable results, which we leave as an open problem for the future.

# Acknowledgments

We thank V. Bulchandani and B. Pozsgay for useful comments on the manuscript. BF is grateful to K. Hódsági for advising him in the early stages of this work.

**Funding information** The work of BF was partially supported by the National Research Development and Innovation Office of Hungary via the scholarship ÚNKP-22-2-II-BME-24. GT was partially supported by the National Research, Development and Innovation Office (NKFIH) through the OTKA Grant K 138606, and also within the Quantum Information National Laboratory of Hungary (Grant No. 2022-2.1.1-NL-2022-00004).

# A   Leading order corrections of the $\phi^4$ and $\phi^6$ THA

Here we follow the approach introduced in Ref. [30]. Let $H$ denote the full Hamiltonian, $\mathcal{E}$ one of its eigenvalues, and $\psi_{\mathcal{E}}$ the corresponding eigenstate:

$$H\psi_{\mathcal{E}} = \mathcal{E}\psi_{\mathcal{E}}. \tag{A.1}$$

The truncation splits the Hilbert space into the low-energy subspace ($\mathcal{H}_l$) retained for the numerical calculations, and the high-energy ($\mathcal{H}_h$) subspace which is discarded. Then the full Hilbert space is $\mathcal{H} = \mathcal{H}_l \oplus \mathcal{H}_h$ and the eigenvalue equation can be decomposed as

$$\begin{pmatrix} H_{ll} & H_{lh} \\ H_{hl} & H_{hh} \end{pmatrix} \begin{pmatrix} \psi_{\mathcal{E},l} \\ \psi_{\mathcal{E},h} \end{pmatrix} = \mathcal{E} \begin{pmatrix} \psi_{\mathcal{E},l} \\ \psi_{\mathcal{E},h} \end{pmatrix}, \tag{A.2}$$

where $H_{ll}$ is the naive truncated Hamiltonian. Assuming that the Hamiltonian has the form $H = H_0 + V$ with a diagonal part $H_0$ and a perturbation $V$ yields

$$(H_{ll} + \Delta H)\psi_{\mathcal{E},l} = \mathcal{E}\psi_{\mathcal{E},l}, \tag{A.3}$$

where

$$\Delta H = -V_{lh}(H_0 + V_{hh} - \mathcal{E})^{-1}V_{hl} = -V_{lh}(H_0 - \mathcal{E})^{-1}V_{hl} + \mathcal{O}(V^3), \tag{A.4}$$

are counter terms that eliminated the cut-off dependence of the energy level. We introduce a computational basis $|k\rangle$ composed of the eigenstates of $H_0$:

$$H_0 |k\rangle = E_k |k\rangle, \tag{A.5}$$

and approximate the counter terms by keeping only the leading order term, which has the matrix elements

$$(\Delta H)_{ij} = -\sum_{k:E_k>\Lambda} \frac{V_{ik}V_{kj}}{E_k - \mathcal{E}} = -\int_{\Lambda}^{\infty} dE \frac{M(E)_{ij}}{E - \mathcal{E}}, \tag{A.6}$$

where

$$M(E)_{ij}dE = \sum_{k:E<E_k<E+dE} V_{ik}V_{kj}. \tag{A.7}$$

Since the relevant contribution comes from the high-energy asymptotics of $M(E)$, it is reasonable to approximate $M(E)$ as a continuous distribution due to the high density of energy levels in $\mathcal{H}_h$. The matrix elements $M(E)_{ij}$ can be evaluated explicitly by introducing the quantity

$$C_{ij}(\tau) = \langle i|V(\tau/2)V(-\tau/2)|j\rangle = \int_0^{\infty} dE\, e^{-(E-(E_i+E_j)/2)\tau} M(E)_{ij}, \tag{A.8}$$

where

$$V(\tau) = e^{H_0\tau}Ve^{-H_0\tau}, \tag{A.9}$$

is the perturbation in (Euclidean) interaction picture. The high-energy behaviour can be determined from the behaviour for small $\tau$, which can be obtained using Laplace-transformation. Assuming that the Hamiltonian has the form

$$H_0 = \frac{1}{2}\int dx : \left((\partial_t\phi)^2 + (\partial_x\phi)^2 m^2\phi^2\right) :,$$

$$V = \sum_n g_n V_n, \qquad V_n = \int dx : \phi^n(x) :, \tag{A.10}$$

and applying Wick's theorem yields

$$\frac{d}{d\tau}C_{ij}(\tau) = \sum_{n,m}g_n g_m \sum_{0\leq k\leq \min(n,m)} k!\binom{n}{k}\binom{m}{k}I_k'(\tau)\langle i|V_{n+m-2k}|j\rangle \tag{A.11}$$

$$= \int_0^\infty dE e^{-(E-(E_i+E_j)/2)\tau}\left[-(E-(E_i+E_j)/2)M(E)\right], \tag{A.12}$$

where

$$I_k(\tau) = \int_{-L/2}^{L/2} G_L(\tau,z)^k, \tag{A.13}$$

and $G_L(\tau,z)$ is the Euclidean propagator

$$G_L(\rho) = \frac{1}{2\pi}K_0(m\rho) \approx -\frac{1}{2\pi}\log\left(\frac{e^\gamma}{2}m\rho\right)\left[1+\mathcal{O}(m^2\rho^2)\right], \qquad \rho m \ll 1. \tag{A.14}$$

Here $\rho = \sqrt{\tau^2+z^2}$ is the Euclidean distance, and $K_0$ is the modified Bessel function of the second kind. The derivative with respect to $\tau$ was introduced to eliminate some spurious IR divergences [30].

We only keep the leading non-analytic behaviour of $I_k'(\tau)$ for $\tau \to 0$, which corresponds to the leading order in the cutoff dependence, expressed as a Laplace transform of some function $\mu_k(E)$

$$I_k'(\tau) = \int_\varepsilon^\infty dE e^{-E\tau}\mu_k(E) + \text{subleading contributions}. \tag{A.15}$$

After substituting back and reordering the sum, the final expression for the correction term is

$$(\Delta H)_{ij} = -\sum_{k'}(V_{k'})_{ij}\sum_{n\geq m}g_n g_m \int_{\Lambda-(E_i+E_j)/2}^\infty dE \frac{\mu_{nmk'}(E)}{E+(E_i+E_j)/2-\mathcal{E}}, \tag{A.16}$$

where

$$\mu_{nmk'}(E) = -(2-\delta_{n,m})k!\binom{n}{k}\binom{m}{k}\frac{\mu_k(E)}{E}\Bigg|_{k=(n+m-k')/2}. \tag{A.17}$$

Neglecting the $E_i + E_j$ terms in comparison with the cut-off $\Lambda$, the result can be simplified further as

$$\Delta H = -\sum_k \kappa_k V_k, \tag{A.18}$$

where

$$\kappa_k = \sum_{n\geq m}g_n g_m \int_\Lambda^\infty dE \frac{\mu_{nmk}(E)}{E-\mathcal{E}}. \tag{A.19}$$

Note that this still contains the exact eigenvalue $\mathcal{E}$ of the level considered. Following [30] this can be replaced by a value which is chosen to lie in the expected range of the energy level(s) which are the objects of the numerical computation, henceforth referred to as *reference energy*.

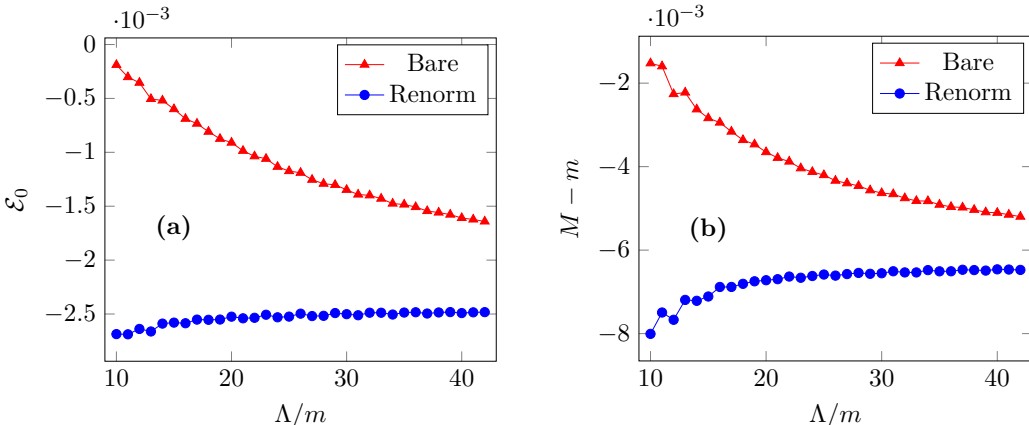

Figure 6: (**a**) The vacuum energy and (**b**) the mass gap ($M = \mathcal{E}_1 - \mathcal{E}_0$)) as a function of the cutoff energy in the $\phi^6$ theory using the naive (*bare*) and the RG-improved (*renorm*) Hamiltonians. Parameters: $mL = 4, \tilde{g}_6 = 0.05$, the reference energy $\mathcal{E}$ was set to 0 for the vacuum energy and $m$ for the mass gap calculation.

The functions $\mu_{nmk}(E)$ relevant for $n, m = 2, 4, 6$ are given by

$$\mu_{220}(E) = \frac{1}{\pi E^2}, \qquad \mu_{422}(E) = \frac{12}{\pi E^2}, \qquad \mu_{440}(E) = \frac{3}{\pi^3 E^2}\left[6(\log E/m)^2 - \frac{\pi^2}{2}\right],$$

$$\mu_{442}(E) = \frac{72 \log E/m}{\pi^2 E^2}, \qquad \mu_{444}(E) = \frac{36}{\pi E^2},$$

$$\mu_{624}(E) = \frac{30}{\pi E^2}, \qquad \mu_{642}(E) = \frac{90}{\pi^3 E^2}\left[6(\log E/m)^2 - \frac{\pi^2}{2}\right],$$

$$\mu_{644}(E) = \frac{720 \log E/m}{\pi^2 E^2}, \qquad \mu_{646}(E) = \frac{180}{\pi E^2},$$

$$\mu_{660}(E) = \frac{675}{2\pi^5 E^2}\left[(\log E/m)^4 - \frac{\pi^2}{2}(\log E/m)^2 + 2\zeta(3)\log E/m + \frac{\pi^4}{80}\right],$$

$$\mu_{662}(E) = \frac{1350}{\pi^4 E^2}\left[2(\log E/m)^3 - \frac{\pi^2}{2}\log E/m + \zeta(3)\right],$$

$$\mu_{664}(E) = \frac{675}{\pi^3 E^2}\left[6(\log E/m)^2 - \frac{\pi^2}{2}\right],$$

$$\mu_{666}(E) = \frac{1800 \log E/m}{\pi^2 E^2}, \qquad \mu_{668}(E) = \frac{225}{\pi E^2}. \tag{A.20}$$

The terms involving $n, m = 2, 4$ agree with those of Ref. [30], while the rest are new results of this work. We can verify the latter by computing the vacuum energy and the mass gap in $\phi^6$ theory at different values of the cutoff, with the results shown in Fig. 6. Comparing the numerical results obtained from the naive truncated (bare) Hamiltonian $H_{ll}$ and those obtained from the renormalised Hamiltonian $H_{ll} + \Delta H$, we find that the counter terms significantly suppress the dependence on $\Lambda$, making THA converge much faster as the cutoff is increased.

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
