# Peer review of "Weak vs. strong breaking of integrability in interacting scalar quantum field theories"

_SciPost Physics, doi:SciPost Phys. 15, 137 (2023)_

## Round 2 · Referee Report · Anonymous (Referee 1) · 2023-7-23

Strengths

1 - original and relevant
2 - up to date and of current interest
3 - the flow of reasoning can be followed very clearly
4 - presentation of results is well documented and understandable
5 - citations of previous literature are correct

Weaknesses

None

Report

The paper deals with the classification of integrability breaking perturbations according to their strength, exploring $\phi^4$ and $\phi^6$ quantum field theories, by methods of Truncated Hamiltonian Approach (THA), an evolution/adaptation of TCSA, of which the authors are experts.
The recently introduced criteria for the classification of integrability breaking make this paper very relevant, current and original.
It is written clearly and the line of reasoning can be followed with ease. The results presented are well complemented by the show of graphs and tables of data, well commented by captions and therefore easy to read and interpret.
The citations to previous literature are, to my knowledge, correct and complete.
For all these reasons, and because the contents fit with the interests of the journal, I strongly support the publication of this paper on SciPost Physics.

Requested changes

None

---

## Round 2 · Referee Report · Anonymous (Referee 2) · 2023-8-7

Strengths

1- The article is well-organized and thoughtfully written. 2- The topic is interesting. 2- The numerical results are convincing.

Weaknesses

1- The justification given on page 7, about the use of the number of particles in the unperturbed theory, as a suitable approximate quantum number to study this crossover phenomena, appears to me slightly misleading.

Report

This article focuses on the characterization of explicit integrability breaking in a 1+1-dimensional quantum field theory associated with a massive free boson perturbed by the operators $\phi^N$, with N=4 or N=6.

The analysis consists in the study of the spectrum of the corresponding Hamiltonian using the truncated space approximation to explore the crossover of the coupling as a function of the number of particles. This is in contrast to the case of spin chains, where the crossover was studied as a function of the volume.
The research topic is interesting, and the article is well-organized. The results have a numerical/phenomenological nature but are quite convincing.

My primary concern centres around the use of the number of particles from the undeformed theory as a "good approximate quantum number" instead of the volume.

This approach seems somewhat risky, as one could argue that the crossover phenomena should be closely tied to the rapid increase in non-elastic processes. From my perspective, the justification presented on page 7—where the authors correctly assert that violating particle number processes are suppressed at low energies—may not be fully applicable in the current context.

In my view, a more thorough justification is needed for solely considering the interacting Hamiltonians within individual subspaces of a given particle number $n$.

Still, the obtained results do indeed appear convincing.

Requested changes

1-Improve/correct the central paragraph on page 7.

2-Provide concrete --comparative-- numerical support for considering the interacting Hamiltonians only within individual subspaces of a given particle number $n$.

---

## Round 3 · Author Response

We are grateful to the referees for the careful reading of our manuscript.

We agree with Referee 2 that our discussion regarding the role of particle numbers was not written with enough care, and we replaced the corresponding text with a more careful argument. We thank the referee for pointing this out. We note further that the whole argument is admittedly heuristic but is justified by our subsequent findings.

---

## Round 3 · List of Changes

2nd and 3rd paragraphs of Section 4 were rewritten to make our point more precise and extended the sentence after Eq. (4.1).

---

## Editorial Decision

published